# Targeting CX3CR1 Suppresses the Fanconi Anemia DNA Repair Pathway and Synergizes with Platinum

**DOI:** 10.3390/cancers13061442

**Published:** 2021-03-22

**Authors:** Jemina Lehto, Anna Huguet Ninou, Dimitrios Chioureas, Jos Jonkers, Nina M. S. Gustafsson

**Affiliations:** 1Science for Life Laboratory, Department of Oncology and Pathology, Karolinska Institute, 171 21 Stockholm, Sweden; jemina.lehto@ki.se (J.L.); anna.huguet@ki.se (A.H.N.); dimitrios.chioureas@ki.se (D.C.); 2Kancera AB, Karolinska Science Park, 171 48 Solna, Sweden; 3Oncode Institute and Division of Molecular Pathology, The Netherlands Cancer Institute, 1066CX Amsterdam, The Netherlands; j.jonkers@nki.nl

**Keywords:** CX3CR1, FANCD2, replication, Fanconi anemia pathway, KAND567

## Abstract

**Simple Summary:**

Chemotherapeutics exerting their antiproliferative actions by introducing DNA crosslinks, such as platinum drugs, are used to treat numerous cancers. Unfortunately, their therapeutic potential is limited due to adverse side effects and acquired resistance, the latter often associated with enhanced DNA repair capacity. Thus, targeting DNA repair is a promising strategy to lower effective doses and associated side effects, and to restore sensitivity to treatment. The C-X3-C motif chemokine receptor 1 (CX3CR1) is an emerging anticancer target which expression correlates with worse overall survival in cancer patients undergoing DNA damaging treatments. Here we show for the first time that the clinical-phase small molecule inhibitor KAND567 targeting CX3CR1 augments the efficacy of DNA crosslinking chemotherapeutics in cancer cell lines, including platinum resistant models, by interference of the Fanconi anemia DNA repair pathway. Hence, the interplay between CX3CR1 and FA repair provides novel potential therapeutic opportunities in cancers treated with DNA crosslinking agents.

**Abstract:**

The C-X3-C motif chemokine receptor 1 (CX3CR1, fractalkine receptor) is associated with neoplastic transformation, inflammation, neurodegenerative diseases and aging, and the small molecule inhibitor KAND567 targeting CX3CR1 (CX3CR1i) is evaluated in clinical trials for acute systemic inflammation upon SARS-CoV-2 infections. Here we identify a hitherto unknown role of CX3CR1 in Fanconi anemia (FA) pathway mediated repair of DNA interstrand crosslinks (ICLs) in replicating cells. FA pathway activation triggers CX3CR1 nuclear localization which facilitates assembly of the key FA protein FANCD2 into foci. Interfering with CX3CR1 function upon ICL-induction results in inability of replicating cells to progress from S phase, replication fork stalling and impaired chromatin recruitment of key FA pathway factors. Consistent with defective FA repair, CX3CR1i results in increased levels of residual cisplatin-DNA adducts and decreased cell survival. Importantly, CX3CR1i synergizes with platinum agents in a nonreversible manner in proliferation assays including platinum resistant models. Taken together, our results reveal an unanticipated interplay between CX3CR1 and the FA pathway and show for the first time that a clinical-phase small molecule inhibitor targeting CX3CR1 might show benefit in improving responses to DNA crosslinking chemotherapeutics.

## 1. Introduction

Chemotherapeutics exerting their action by forming DNA crosslinks, such as mitomycin C and the platinum analogues cisplatin and carboplatin, are widely used drugs within oncology. Among the different types of DNA crosslinks introduced, the covalent links between complementary strands (i.e., DNA interstrand crosslinks, ICLs) are considered the most toxic ones as they prevent strand separation ultimately resulting in chromosomal breakage if left unrepaired [1]. Replication forks that stall at the ICLs are immediately recognized by the Fanconi anemia (FA) pathway which stabilizes the fork and mediates ICL repair and replication in order to preserve genome stability. Mutations in any of the 22 FA genes identified to date result in FA, a rare genetic disease characterized by bone marrow failure, predisposition to cancer and hypersensitivity to ICL inducing agents [2]. In addition, FA pathway activation contributes to acquired cisplatin resistance [3,4,5,6], a common clinical problem. Altogether, these highlight the importance of a functional FA pathway.

Central in the FA pathway is FANCD2. Together with FANCI, FANCD2 forms the I-D complex which associates to the stalled fork where FANCD2 becomes monoubiquitinated by the FA core complex [7], which can be visualized as FANCD2 nuclear foci at the ICLs and constitute a hallmark of FA pathway activation. The critical functions of the I-D complex in FA repair include serving as a recruitment platform for downstream repair factors and protecting the ICL-stalled replication forks. Fork protection is achieved by filament formation on the DNA as well as by interacting with and stabilizing RAD51-DNA filament complexes [2,8,9,10]. RAD51 is recruited within minutes upon ICL fork stalling to mediate fork remodeling and reversal, distinct from its function in homologous recombination (HR) repair of DNA double-strand breaks (DSBs) [8,10,11,12]. In addition to fork protection, FANCD2 regulates the nucleolytic incision step through recruitment of nucleases, leading to ICL unhooking and generation of DSBs [2,13]. The opposite strand of the unhooked ICL undergoes translesion synthesis and the DSB intermediates are repaired preferably via HR [2].

The chemokine (C-X3-C) motif receptor 1 (CX3CR1) is associated with ovarian, breast, prostate, colorectal, testicular, pancreatic and gastric cancer, B cell malignancies, and glioblastoma [14]. In cancer cells, CX3CR1 drives invasion, metastasis, proliferation, and survival [14]. In addition, the receptor is also associated with inflammation, neurodegenerative diseases [15], and aging [16], and the CX3CR1 inhibitor KAND567 [17] is under clinical trials for hyperinflammatory conditions (EudraCT: 2020-002322-85). Notably, only ten percent of CX3CR1-expressing cells migrate to its soluble fractalkine ligand [18], suggesting that the mechanisms behind CX3CR1-driven cancer proliferation goes beyond ligand-mediated chemotaxis. In support of this notion, CX3CR1 has been identified in genome-wide siRNA screens aiming to identify regulators of genome stability [19] and its knockdown sensitizes ovarian cancer xenografts to DNA-damaging ionizing radiation. Furthermore, CX3CR1 mRNA expression correlates with worse overall survival for patients undergoing treatment with DNA damaging chemotherapeutics [20], altogether pointing towards a potential function of CX3CR1 in maintenance of genome stability. However, how CX3CR1 contributes to genome stability remains poorly understood.

Here we report a previously unknown role of CX3CR1 in FA-mediated DNA repair and evaluate for the first time the in vitro efficacy of targeting CX3CR1 with a small molecule inhibitor in the context of cancer. CX3CR1 inhibition synergizes with cisplatin and carboplatin in viability assays with an improved synergistic effect in platinum resistant cells. We show that upon ICL induction, CX3CR1 inhibition results in replication fork stalling, increased levels of unresolved DNA-cisplatin adducts and inability of replicating cells to progress from S phase. Mechanistically, CX3CR1 inhibition reduces FANCD2 foci formation and chromatin recruitment, blocking downstream events in the FA pathway and resulting in defective ICL repair.

## 2. Materials and Methods

### 2.1. Cell Lines

A2780 (#93112519) and A2780Cis (#93112517) cell lines were purchased from European Collection of Authenticated Cell Cultures (ECACC, Sigma-Aldrich, St. Louis, MO, USA). VH10, U2OS, SKOV-3, and HEK293T cell lines were purchased from American Type Culture Collection (ATCC^®^, Manassas, VA, USA). The PEO1 and PEO1.C2-4 were a gift from Dr. Kumar Sanjiv (cell line identity had been confirmed by Sanger sequencing), the BJhTERT cells were provided by W. Hahn (Dana–Farber Cancer Institute) and OVCAR-4 cells were a kind gift from Dr. Kaisa Lehti. A2780, A2780cis, PEO1, PEO1.C2-4, and OVCAR-4 were cultured in RPMI 1640 with GlutaMAX (#61870 Thermo Fisher Scientific, Waltham, MA, USA) with supplementation of 2% pyruvate (#11360070 Thermo Fisher, for PEO1 and PEO1.C2-4). U2OS were cultured in DMEM with GlutaMAX (#31966-021, Thermo Fisher Scientific), HEK293T cells were cultured in Iscove’s Modified Dulbecco’s Medium (IMDM; #12440, Thermo Fisher Scientific) and SKOV-3 cells were cultured in McCoy 5A (#16600082, Thermo Fisher Scientific). All media was supplemented with 10% fetal bovine serum (FBS; #10500, Thermo Fisher Scientific), except for U2OS with 5% FBS, and 100 U/mL penicillin– streptomycin antibiotics (#15070, Thermo Fisher Scientific). Cell were grown at 37 °C in 5% CO_2_ humidified incubators.

### 2.2. Cloning and Production of Lentiviral Particles

Hairpins targeting CX3CR1 were derived from Sigma Mission library (TRCMm1.0) [21]. Competent *E. coli* retrieved from glycerol stocks were let to expand for 16 h in Luria–Bertani (LB) medium with 100 µg/mL ampicillin (Sigma-Aldrich, #A5354) at 37 °C and 250 rpm. Plasmids were isolated using PureLink™ HiPure Plasmid Midiprep Kit (Invitrogen #K2100, Carlsbad, CA, USA). Seven million HEK293T cells were seeded 24 h prior lentiviral particle transfection on 150 mm dishes. As described before [22], plasmid containing shRNA and lentiviral packaging vectors (with VSV-G envelope) were transfected into HEK239T cells dropwise using calcium chloride (Sigma-Aldrich, #C1016) to permeabilize the cells. Iscove’s Modified Dulbecco’s Medium (#12440061, Thermo Fisher Scientific) was changed 16 h post particle transfection and cells were let to expand for 24 h until produced lentivirus was collected from the supernatant through 0.2 µm filter and concentrated for 2 h at 22,000× *g*. Lentivirus amount was quantified by RT-qPCR (Applied Biological Materials; qPCR Lentivirus Titration Kit) according to manufacturer’s protocol; shRNA sequences used are in Appendix A.

### 2.3. Generation of shRNA Cell Lines

Twenty-four hours prior transfection, 1 × 10^5^ cells were seeded into 100 mm plates. The following day, media were renewed and lentiviral particles were transduced with multiplicity of infection (MOI) 25 with 8 μg/mL polybrene (#TR-1003-G, Sigma-Aldrich). At 24 h post-transduction, media were changed to media containing 3 mg/mL puromycin (#A1113803, Thermo Fisher Scientific) and selection was performed for 72 h or until control cells were dead. Successful knockdown was verified using quantitative RT-PCR and Western blot.

### 2.4. Drug Synergy Studies

Compounds were spotted using Tecan D300 dispenser (Tecan, HP Inc, CA, USA) on 384-well plates, 0.3% Tween-20 was added as a surfactant to H_2_O-diluted platinum compounds. Cells were dispensed using Multidrop Combi (Thermo Scientific) dispenser at a density of 1300–1600 cells/well, incubated for 72 h, and resazurin sodium salt (100 µg/mL) was added. Fluorescence intensity at 544/590 nm (ex./em.) was measured with Hidex Sense (Hidex Oy) microplate reader. Dose-response matrix viability values for two compounds were used to calculate delta scores using the Zero Interaction Potency (ZIP) model with Synergy Finder (https://synergyfinder.fimm.fi/, last accessed on 15 March 2021) [23].

### 2.5. Colony Formation Assays

Two hundred to two thousand (Appendix A) cells were seeded per well in 6-well plates 24 h prior indicated treatments. Post-treatment, cells were washed in PBS (#14190, Thermo Fisher Scientific) and released into fresh media with 10% FBS. Cells were let to grow ~14 days, media were removed, the wells were washed in PBS carefully, and the colonies were stained with 4% methylene blue (#M9140, Sigma-Aldrich, Sweden) in methanol for 20 min. The colonies were manually counted and relative colony count to DMSO control was calculated. Alternatively, colonies were quantified prior to fixation with resazurin sodium salt (100 µg/mL), fixed with 4% formaldehyde (#281692, Santa Cruz Biotechnology, Dallas, TX, USA) in PBS and stained with crystal violet solution (#C0775, Sigma-Aldrich).

### 2.6. Proliferation Assays for Live-Cell Analysis

Indicated cell lines were seeded into 96-well plates, treated, and the plates were placed into the IncuCyte system (Incucyte^®^ S3; Essen Bioscience, Inc., Ann Arbor, MI, USA) with real-time imaging every 4 h (4 images per well) for 168 h. Label-free cell confluency (% confluence) was calculated by Incucyte^®^ Live-Cell Analysis System Module as the occupied area with cells versus empty area of cell images acquired over time. Growth curves were constructed using calculated confluency values for each time point and proliferation curves were created with GraphPadPrism software (GraphPad Software Company, San Diego, CA, USA).

### 2.7. Flow Cytometry

For cell cycle analyses of replicative cells, cells were synchronized with 6 µM aphidicolin for 24 h, and released in media with 10 µM 5-Ethynyl-2′-deoxyuridine (EdU) (Sigma-Aldrich, #1T511285) for 45 min, washed in PBS, and then treated for 6 or 16 h with indicated treatments. Cells were collected and fixed in 70% ice-cold ethanol in PBS followed by permeabilization in 0.1% saponin (Sigma-Aldrich, #47036) and 1% BSA (Sigma-Aldrich, #A9418) in PBS for 30 min on ice. Click iT master mix was prepared as follows and pipetted in order: 42.55 µL PBS, 2 µL CuSO_4_ (Honeywell Riedel-de Haën, #12849, Seelze, Germany), 0.3 µL ATTO 647 azide (ATTO-TEC, #AD 647-101, Siegen, Germany) (2 mM), 5 µL ascorbic acid (Sigma-Aldrich, #A0278; 100 mM), and cells were incubated for 30 min covered from light. Cells were resuspended into 1% BSA (Sigma-Aldrich, #A9418) in PBS and washed twice for 5 min, at 5000× *g*, at 4 °C and then resuspended in DNA staining buffer containing 40 μg/mL propidium iodide (Sigma-Aldrich #P4170), 100 μg/mL RNase A (Thermo Scientific, #EN0531), 0.1% Triton X-100 (Sigma-Aldrich, #T8787) in PBS and incubated for 20 min in the dark. Assessment of cisplatin-DNA adducts in flow cytometry was performed as previously described [24] with the following modification: cell permeabilization and antibody incubations were performed in 0.1% saponin (Sigma-Aldrich, #47036) and 1% BSA (Sigma-Aldrich, #A9418) in PBS for 30 min on ice. DNA was stained with 10 μg/mL Hoechst (Invitrogen, #H3570) together with 100 μg/mL RNase A (Thermo Scientific, #EN0531), 0.1% Triton X-100 (Sigma-Aldrich, #T8787) in PBS. Samples were analyzed by flow cytometry (FACS Navios; Beckman Coulter, Brea, CA, USA) and quantified by Kaluza software, 20,000 events per sample were gated. Antibodies used are listed in Appendix A.

### 2.8. DNA Fiber Assay

DNA fiber assay was performed as previously described [25] with the following modification: 1.5 × 10^5^ A2780 and A2780Cis cells were seeded into 6-well plates. The following day cells were synchronized at the G1/S boundary by 6 µM aphidicolin for 24 h [26]. Cells were washed 2 times with warm PBS, one time with warm RPMI media, and treated as indicated. During the last hour of drug treatment, cells were labeled with CldU 25 μM (Sigma-Aldrich, C6891) for 30 min and then with 250 μM IdU (Sigma-Aldrich, #I7125) for 30 min. Cells were harvested by trypsinization, resuspended in PBS, and samples were diluted to approximately 7 × 10^5^ cells/mL. Fiber spreading, denaturation, fixation, and staining were performed in ibiTreat µ-slide VI 0.4 (Ibidi, #80606). Fluorescence images were captured using a NikonTi2 fluorescence microscope with the 40× objective (with MilliQ water) with excitation wavelengths of 488 and 568 nm, and analyzed using the ImageJ software. At least 100 unidirectional forks labeled with both CldU and IdU were measured for every condition using Fiji [27,28]. Antibodies used are listed in Appendix A.

### 2.9. Confocal Microscopy

Cells were seeded on coverslips (VWR #631-1580) in 12-well plates and let to attach overnight before treatments. Prior fixation, cells were washed with PBS (Gibco, #14190) and treated with CSK buffer (10 mM PIPES (Sigma-Alrdich, #P6757), 100 mM NaCI (EMD Millipore, #31434-M), 300 mM sucrose (Sigma-Aldrich, #S0389), 3 mM MgCl2 (Sigma-Aldrich, #M9272), 0.7% TritonX-100 (Sigma-Aldrich, #T8787) for 2 min to remove soluble proteins, washed with PBS and fixed with 4% formaldehyde (#281692, Santa Cruz Biotechnology) in PBS containing 2% sucrose (Sigma-Aldrich, #S0389) for 10 min. Cells were permeabilized with 0.2% NP-40 (BioVision, #2111, 10% in water) in PBS for 10 min and washed gently with PBS. Blocking was performed for 30 min with 2% BSA (Sigma-Aldrich, #A9418), 5% glycerol (Sigma-Aldrich, #49781), 0.1% Tween-20 (Sigma-Aldrich, #P1379) in PBS. Cells were washed, primary antibodies diluted in blocking solution was added to the coverslips let at 4 °C overnight. The following day, coverslips were washed with PBS and incubated with secondary antibodies dilutes in blocking solution for 45 min at RT, washed with PBS, and incubated with DAPI (Sigma-Aldrich, #D9542) for 10 min in RT protected from light and washed with PBS. Coverslips were mounted on microscope slides (VWR, #631-1554) with ProlongGold (#P36934, Thermo Fisher Scientific) and let to set for 24 h in 4 °C in dark. Images were acquired with Zeiss LMS 780 confocal microscope (Carl Zeiss, Thornwood, NY) and analyzed using CellProfiler software, www.cellprofiler.org, last accessed on 15 March 2021 [29]. Antibodies used are listed in Appendix A.

### 2.10. Fractionation of Soluble and Chromatin-Bound Proteins

Cells were seeded on 100 mm plates and the following day treated as indicated. Cells were harvested by trypsination (Trypsin, Life Technologies, #15400-054), spun down at 300 G for 5 min to pellet cells, resuspended in 1 mL ice-cold PBS and counted. Equal amount of cells were pelleted at 300 G for 5 min and resuspended in 70 µL low-salt buffer (LSB; 10 mM HEPES (Sigma-Aldrich, #H3375) pH 7.4, 10 mM potassium chloride (KCl, Sigma-Aldrich, #P9541) and 0.05% NP40, (BioVision, #2111) supplemented with protease (Roche, #04693159001) and phosphatase (Thermo Fisher Scientific, #78426) inhibitors. Samples were incubated on ice for 5 min, vortexed briefly and centrifuged at 3600× *g* for 5 min at 4 °C for separation of soluble cell fraction (supernatant) and chromatin fraction (cell pellet). Supernatant containing the soluble cell fraction was centrifuged at 13,000× *g* for 10 min to collect debris and insoluble proteins Pellet containing the chromatin fraction was washed twice in LSB, spun down at 3000× *g* for 5 min at 4 °C. Chromatin-bound extraction buffer was prepared by adding 2.5 µL 10× MNase buffer (CaCl_2_), 0.25 µL 100× BSA, and 1 µL Micrococcal Nuclease (New England Biolabs, #M0247S) to 25 µL nuclease buffer (150 mM NaCl, 5 mM MgCl_2_) in RT. Pellets were resuspended in nuclease extraction buffer, vortexed for 15 s, incubated for 10 min at 37 °C, and vortexed for 15 s, and finally centrifuged at 13,000× *g* for 10 min. Twenty-six microliters of each supernatant containing the chromatin-bound nuclear fraction or soluble fraction was mixed with 4 µL 10× (#NP0009, Thermo Fisher Scientific) and 10 µL NuPage LDS sample buffer 4× (#NP0007, Thermo Fisher Scientific), incubated at 70 °C for 10 min, spun down quickly, and loaded on SDS-PAGE gels (BioRad, #4561085).

### 2.11. siRNA Transfections

Cells were seeded on 6-well plates at a density of 200,000 per well and let to attach overnight. CX3CR1 siRNAs or siRNA nontargeting control (Qiagen #1027281) was mixed with 200 µL of serum free medium to the final concentration of 20 nM. 12 µL of interferin (Source Bioscience, #409-50) was added to Eppendorf tubes and were vortexed for 15 s. Tubes were incubated in RT for 15 min and 200 µL was added to the cells into 6 wells plates containing 2 mL media. Cells were incubated with siRNAs for 72 h. For confocal studies, washout and replacement of fresh medium was performed 6 h post transfection and cells were trypzinized and reseeded on confocal slides on 12 well plates 30 h post transfection. Synchronizations with aphidicolin were performed 48 h post transfection for 24 h. siRNA sequences used are in Appendix A.

### 2.12. Immunoblotting

Samples were loaded into Mini-Protean precast gels (BioRad #4561085) and run with 130 V in Tris/Glycine/SDS TGS buffer (BioRad, #161-0772). Samples were transferred to nitrocellulose membranes (BioRad, #1704270/1704271) using TransBlot Turbo transfer system (BioRad). Successful transferred was checked with Ponceau staining (Sigma #P7170) and the membranes were washed in TBS-T (Cell Signaling, #9997) before blocking in Odyssey Blocking Buffer (LiCor, #927-50000)/TBS-T (1:1) for 1 h. Incubation with primary antibodies diluted in TBS-T/Odyssey blocking buffer was done overnight at 4 °C or for 30 min–2 h at RT. Membranes were washed in TBS-T and incubated with secondary antibodies for 1 h at RT, and developed using Odyssey Fc Imager (LI-COR Biosciences) and analyzed with Image Studio software (LI-COR Biosciences). Antibodies used are listed in Appendix A.

### 2.13. Quantitative Real-Time PCR (qRT-PCR) Analysis

For the RT-qPCR analysis, RNA was isolated using a DirectZol RNA extraction kit (ZymoResearch, #R2051) following manufacturer’s instructions and cDNA was synthetized by RT reaction using QuantiTect cDNA synthesis kit (Qiagen, #205311). qRT-PCR reaction was performed with Luminars HiGreen qPCR kit (Thermo Scientific, #K0391) for quantification of *CX3CR1* expression (forward: CACAAAGGAGCAGGCATGGAAG, *reverse: CAGGTTCTC*TGTAGACACAAGGC) using *18S* (forward: AGTCCCTGCCCTTTGTACACA, reverse: GATCCGAGGGCCTCACTAAAC), *β-actin* (forward: CCTGGCACCCAGCACAAT, reverse: GGGCCGGACTCGTCATACT) and *HPRT* (forward: GACCAGTCAACAGGGGACAT, reverse: AACACTTCGTGGGGTCCTTTTC) as housekeeping genes. Relative mRNA levels were calculated in reference to the housekeeping genes using the 2^−ΔΔ*C*T^ method [30].

### 2.14. Treatment with Small Molecule-Inhibitors

Mitomycin C (MMC) (Santa Cruz Biotechnology, #sc-3514B) was stocked in filter sterilized aqueous solution. Cisplatin (CDDP; Sigma-Aldrich, #P4394) and carboplatin (CBP; SelleckChem, #S1215) were dissolved in 0.9% NaCl H_2_O or PBS. Aphidicolin (Sigma-Aldrich, #A0781) and CX3CR1 inhibitor KAND567 (Kancera AB, Solna, Sweden) were dissolved in Dimethyl sulfoxide (DMSO; Sigma-Aldrich, #41369) and DMSO control was used in experiments at the same volume as the volume of highest inhibitor volume. Media were supplemented with 5% FBS (Gibco, 10500064#) in experimental settings involving small molecule inhibitors.

### 2.15. Statistical Analysis

Statistical analyses were performed using GraphPad Prism software. Multiple comparisons were performed with one-way ANOVA or two-way ANOVA followed by Sidak’s, Dunnett’s, or Tukey’s multiple comparisons test. Data are presented as means ± SD or SEM with 95% confidence interval (* *p* < 0.05, ** *p*  <  0.01, *** *p*  <  0.001).

## 3. Results

### 3.1. CX3CR1i Reverses Platinum Resistance

To assess the potential benefit of combining inhibition of CX3CR1 together with platinum drugs, we first performed drug synergy studies [23] using a cell-based assay with viability as a readout. The combination treatments revealed synergistic delta scores between the CX3CR1 inhibitor KAND567 [17], from here on referred to as CX3CR1i, and the DNA-damaging platinum drugs carboplatin and cisplatin in all cancer cell lines tested, including the isogenic cancer cell lines sensitive (PEO1, A2780) and resistant (PEO1.C2-4, A2780Cis) to platinum drugs (Figure 1A and Appendix A). Importantly, drug concentrations resulting in strong synergistic effects on viability in cancer cell lines displayed a weak synergistic effect in nontransformed cells (VH10, BJhTERT), indicating a potential poor cytotoxicity in normal, healthy cells at doses effective in cancer cells. Notably, the synergy scores where significantly higher in the resistant (PEO1.C2-4, A2780Cis, SKOV3) than sensitive (PEO1, A2780, OVCAR-4) cancer cell lines, indicating a greater synergistic effect.

We next assessed the reversibility of the synergies upon the combination treatment in long-term survival assays by pretreating the cells with CX3CR1i prior to carboplatin treatment followed by drug washout. The synergy with carboplatin was nonreversible in a dose-dependent manner at 14 days post removal of the drugs (Figure 1B). The chemosensitization to platinum upon loss of CX3CR1 activity was confirmed in shCX3CR1 cells upon live-cell imaging of cell growth and in clonogenic survival assays with drug washout (Figure 1C,D and Appendix A) [20]. Altogether, inhibition of CX3CR1 and platinum synergize irreversibly in blocking proliferation, including platinum resistant models.

### 3.2. CX3CR1 Promotes Replicative S-Phase Repair of DNA Crosslinks

On the basis of the strong synergistic effect between CX3CR1i and platinum, we next assessed the role of CX3CR1 in the cellular response to DNA crosslinks. Since the response to platinum drugs is closely related to DNA repair capacity and FA pathway activation contributes to acquired cisplatin resistance [3,5,6], we reasoned that potential defects in replicative FA repair upon CX3CR1i should delay progression of replicating cells.

To address the impact of CX3CR1i on S phase progression upon DNA crosslink induction, we synchronized the A2780 and A2780Cis cells at the G1/S boundary by aphidicolin treatments [26] followed by washout and pulse labeling of replicating cells by the thymidine analogue 5-Ethynyl-2′-deoxyuridine (EdU) prior indicated drug treatments (Figure 2A). Since cisplatin and carboplatin generate ICLs to the same extent but cisplatin is more reactive [31], we from hereon used cisplatin for mechanistic studies.

Monitoring the cell cycle progression of A2780 cells that were replicating at the time of the drug treatments revealed prolonged accumulation of cells in the S phase upon CX3CR1i and cisplatin combination treatments while cisplatin single treated cells progressed into G2/M over time (Figure 2A,B and Appendix A). Consistent with A2780Cis cells being resistant to platinum treatments, cisplatin alone did not influence the progression of the replicating cells (Figure 2A,B and Appendix A). Notably, the CX3CR1i and cisplatin combination treatment in A2780Cis cells resulted in an inability of replicating cells to progress from S phase at 6 h (27% increase) which was largely maintained at 16 h (21%) compared to vehicle and single drug treatments. While cisplatin treated A2780Cis cells were able to proceed into G0/G1 phase at 16 h post treatment, the combination-treated cells failed to proceed to the next generation G1 (Figure 2A,B), indicative of inability for replicating cells to progress from S phase due to delayed or defective DNA repair. Notably, this resembles the phenotype of FA cells which accumulate at late S phase upon ICL induction due to their deficiency in ICL repair [13,32].

Given the failure of replicating cells to progress from S phase, we reasoned that defective FA repair by CX3CR1i treatment should further increase ICL-induced replication fork stalling. In support of this hypothesis, single-molecule analysis of DNA replication dynamics revealed that short-term (3 h) CX3CR1i and cisplatin combination treatments in synchronized cells drastically reduced replication fork progression and shortened replication track lengths in comparison to vehicle and cisplatin single treatments in A2780 (Figure 2C,D). Whilst A2780C is cells treated with cisplatin single agent showed mild effects in replication fork stalling, cotreatment with cisplatin and CX3CR1i resulted in significant decrease of both fork speed and track length comparable to the decline seen in A2780 platinum sensitive cells (Figure 2C,D). In addition, the percentage of cisplatin-DNA adducts significantly increased at 24 h of combination treatment (Figure 2E). Collectively, these results indicate that CX3CR1 participates in resolving DNA crosslinks during S phase.

### 3.3. CX3CR1 Accumulates in the Nucleus upon FA Pathway Activation

To further investigate a potential role for CX3CR1 in FA repair, CX3CR1 localization was assessed upon FA pathway activating treatments with low dose hydroxyurea, cisplatin and mitomycin C for 24 h [33]. Since the human U2OS cells is a well-known model system to assess DNA repair foci formation we employed this cell line for further confocal analyses. Notably, a nuclear accumulation of CX3CR1 occurred to similar extent as the key FA protein FANCD2 in conditions activating the FA pathway (Figure 3A,B). We also identified an increased nuclear localization of CX3CR1 upon 3 h of cisplatin and mitomycin C treatments during S phase which was abolished upon CX3CR1 knockdown, confirming the specificity of the staining (Figure 3C,D and Appendix A). Hence, activation of the FA pathway promotes CX3CR1 nuclear localization.

### 3.4. CX3CR1 Is Required for Recruitment of FANCD2

We identified a strong correlation between FA-induced CX3CR1 and FANCD2 levels upon hydroxyurea, cisplatin and mitomycin C (Figure 4A). This, together with the impaired resolution of DNA crosslinks in replicating cells upon CX3CR1i and the central role of FANCD2 foci formation for functional ICL repair [2], sought us to investigate whether CX3CR1 modulates the recruitment of FANCD2. Since the FA pathway primarily mediates replication-dependent ICL repair [13], cells were synchronized at the G1/S boundary by aphidicolin followed by three hours release to allow for progression into mid-S phase. Blocking CX3CR1 with siRNA or CX3CR1i abolished the cisplatin- and mitomycin C-induced FANCD2 foci formation (Figure 4B,C and Appendix A) as well as FANCD2 chromatin recruitment upon mitomycin C treatments (Figure 4D and Appendix A). Notably, CX3CR1 was identified in both the soluble and chromatin fraction, with an increase upon MMC treatments that was abolished upon adding CX3CR1i. Moreover, CX3CR1i decreased the mitomycin C-induced recruitment of the FANCD2 interacting partner FANCI [2] and RAD51 (Figure 4D and Appendix A), consistent with RAD51 interacting with the I-D complex [10]. Furthermore, CX3CR1i blocked mitomycin C-induced phosphorylation of the recognized DSB marker phosphorylated H2AX S139 (γH2AX) (Figure 4D and Appendix A) [34], indicating a lack of FA-mediated DSB induction [2].

## 4. Discussion

Here, we report that the emerging anticancer target CX3CR1 is an integral factor of the FA DNA repair pathway and CX3CR1i provides a novel opportunity to manipulate FA repair and reverse platinum resistance, a recognized clinical challenge.

Central in the FA pathway is the recruitment of FANCD2, although its functional role in this repair pathway is clear, less is known about regulation of its recruitment. Under FA pathway activating treatments [33], nuclear CX3CR1 levels significantly increased and showed a tight correlation with induction of FANCD2 [2] (Figure 3A,B and Figure 4A). The potential relationship between the proteins, as indicated by the correlation coefficients, was confirmed upon loss of CX3CR1 function by gene silencing or inhibition, which impaired ICL-induced recruitment of FANCD2 to the chromatin as well as assembly of FANCD2 into foci (Figure 4B–D), thus CX3CR1 modulates FANCD2 dynamics at the chromatin. In addition, the FANCD2 dimer partner FANCI displayed reduced chromatin association, as well as RAD51, upon CX3CR1 inhibition (Figure 4D), consistent with RAD51 being an interacting protein of the I-D complex [10]. Since chromatin binding of the I-D complex is required for the subsequent recruitment of nucleases [2,13], our findings place CX3CR1 upstream of the incision step in the FA pathway. In support of this, CX3CR1i blocked mitomycin C-induced phosphorylation of the recognized DSB marker γH2AX (Figure 4D) [34], indicating a lack of nucleolytic cleavage of the stalled fork, block of subsequent ICL unhooking, and DSB generation [2]. Ultimately, it resulted in increased levels of unresolved cisplatin-DNA complexes (Figure 2E). Thus, during FA pathway repair, CX3CR1 acts upstream of the generation of DSB intermediates, distinct from its previously described role in DSB repair upon ionizing radiation [20].

In addition to a potential lack of nucleolytic cleavage, combining cisplatin with CX3CR1i increased fork stalling and decreased replication track lengths in both platinum sensitive and resistant cells (Figure 2C,D), indicating reduced fork stability. One possibility is that CX3CR1 promotes a fork protective mechanism through stimulating recruitment of or stabilizing FANCD2, FANCI, and RAD51 (Figure 4B–D) nucleofilaments at the ICL-stalled fork to prevent fork collapse and protect from nucleolytic degradation [8,9,10,35]. Thus, loss of CX3CR1 could result in improper DNA resection at the stalled replication forks, ultimately leading to chromosomal instability. In line with the replication impairments (Figure 2C,D), inhibition of CX3CR1 severely impaired the progression of replicating cells through S phase (Figure 2A,B), a hallmark of defective FA repair [13,32], consistent with the loss of FANCD2 foci (Figure 3A,B), and further supported by the increased levels of cisplatin-DNA complexes (Figure 2E).

In consonance with this novel role of CX3CR1 in supporting FA DNA repair, we identified strong synergies between CX3CR1i and platinum in cell proliferation assays in a panel of cancer cell lines (Figure 1A,B). The drug synergies were validated using shRNA mediated knockdown of CX3CR1 both upon drug washout in long-term colony formation and live-cell proliferation assays (Figure 1C,D). This is in good agreement with a previous study showing that transient knockdown of CX3CR1 results in sensitization to cisplatin and carboplatin in the platinum sensitive OVCAR-4 cell line [20]. We further show a dependency on CX3CR1 for cell proliferation of platinum resistant cells treated with platinum-based agents (Figure 1 and Figure 2). Notably, whereas cisplatin and CX3CR1i did not alter cell cycle dynamics of replicating A2780Cis cells when administrated as single agents, the combination treatment resulted in sustained S phase arrest (Figure 2A,B). Furthermore, the moderate reduction on fork speed triggered by cisplatin treatment in A2780Cis indicates an enhanced capacity of resistant cells to cope with replication stress, yet inhibition of CX3CR1i together with cisplatin induced forks stalling to levels comparable to that of sensitive cells (Figure 2C,D). Consistent with a dependency on CX3CR1 for resolution of replication stress upon DNA crosslinks in platinum resistant cells via the FA pathway, FA pathway activation contributes to acquired cisplatin resistance and interfering with the FA pathway sensitizes tumor cells to cisplatin [3,4,5,6,36].

Targeting FA pathway to induce DNA damage repair deficiency is a potential therapeutic strategy to sensitize cancers which are refractory to DNA-damaging agents. In this regard, elevated FA pathway expression and function have been shown to convey broad chemoresistance beyond platinum drugs; for instance, to alkylating agents used in gliomas [37,38] and multiple myeloma [39] and FA pathway deficiency defines sensitivity to topoisomerase inhibitors [40,41]. Similar to platinum drugs, these agents interfere with DNA replication and elicit a high degree of replication stress in highly proliferative cells leading to genomic instability. While there are no specific FA targeting inhibitors currently accepted in the clinic [38], targeting CX3CR1 provides the opportunity to render resistant cells, harboring enhanced FA pathway activation, sensitive not only to platinum, but potentially also to other DNA-damaging agents. Interestingly, CX3CR1 overexpression has been associated to worse overall and reduced progression-free survival of platinum-, gemcitabine-, and topotecan-treated ovarian cancer patients [20], highlighting a prospective therapeutic strategy to restore sensitization by targeting CX3CR1.

## 5. Conclusions

In summary, our study uncovers a novel role for CX3CR1 in modulating the FA repair pathway by enabling FANCD2 recruitment at the stalled replication forks upon ICL induction. Our results imply that CX3CR1 can be localized into nucleus upon FA activating conditions to support replicative ICL repair. We reveal that inhibiting CX3CR1 results in sensitization to platinum drugs, displayed by replication fork stalling, impaired cell cycle progression, unresolved cisplatin-DNA adducts, and loss of cancer cell viability upon platinum treatment. Although further studies are required, given that the CX3CR1i KAND567 is well-tolerated and has passed clinical trials with healthy volunteers, the findings presented here could have implications for the design of future clinical trials.

## Figures and Tables

**Figure 1 cancers-13-01442-f001:**
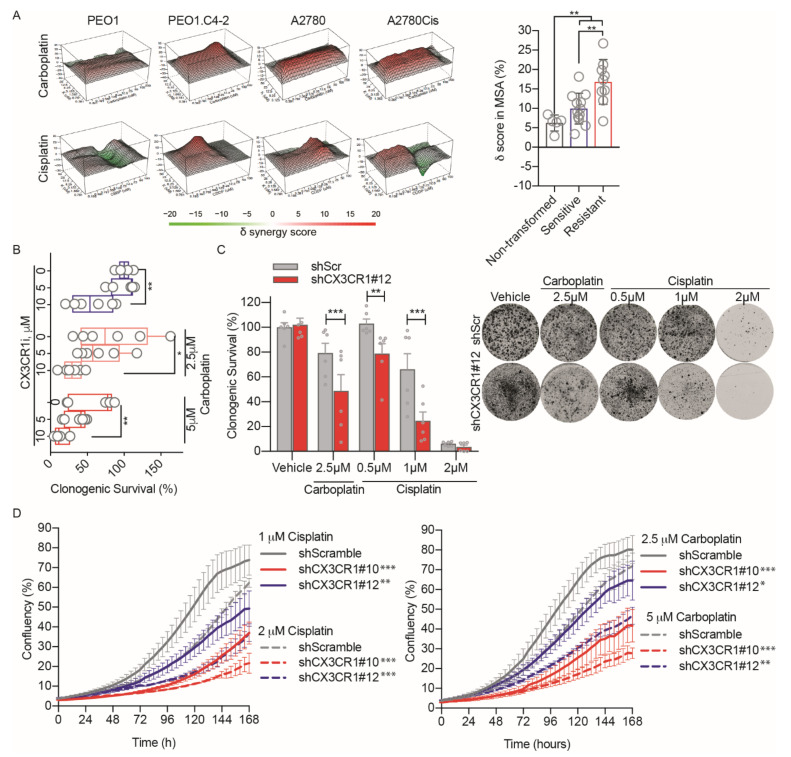
CX3CR1i synergizes with platinum treatments. (**A**) To the left, representative 3D synergy maps for CX3CR1i in combination with carboplatin or cisplatin in isogenic cell lines sensitive (PEO1, A2780) and resistant (PEO1.C2-4, A2780Cis) to platinum drugs. Viability was evaluated at 72 h by resazurin assays and synergy calculated by the Zero Interaction Potency model. To the right, combined average delta scores in most synergistic area based on Zero Interaction Potency model displayed as means ± SD in nontransformed cells (VH10, BJhTERT), platinum sensitive (PEO1, A2780, OVCAR-4), and resistant cell lines (PEO1.C2-4, A2780Cis, SKOV3) upon cisplatin and carboplatin together with CX3CR1i, *n* = 2. Images were generated by Synergy Finder. Statistical significance is compared to sensitive cell lines, ** *p*  <  0.01; two-way ANOVA, Mann–Whitney test. (**B**) Clonogenic survival of A2780 cells treated with CX3CR1i for 24 h, followed by cotreatment for 48 h with carboplatin and then drug wash-out. After 14 days, colonies were stained and manually counted, data displayed as Tukey box plot, *n* = 6; * *p*  < 0.05, ** *p* <  0.01; one-way ANOVA, Dunnett’s multiple comparisons test. (**C**) Colony formation assay of shCX3CR1 or shControl A2780 cells treated with indicated doses of cisplatin or carboplatin for 7 days and then released by washout, let to grow for 10 days, and then fixed and stained with crystal violet (left) with quantification (right). Statistical significance is compared to shControl per each concentration, ** *p*  <  0.01, *** *p*  <  0.001; two-way ANOVA, Šídák’s multiple comparisons test. (**D**) Growth curves (0–168 h) based on the area of confluency (%) of indicated cell lines upon cisplatin (left) and carboplatin (right) treatments. Data displayed as means ± SD, *n* = 3. Statistical significance is compared to shControl, ** *p*  <  0.01, *** *p*  <  0.001; two-way ANOVA, Dunnett’s multiple comparisons test.

**Figure 2 cancers-13-01442-f002:**
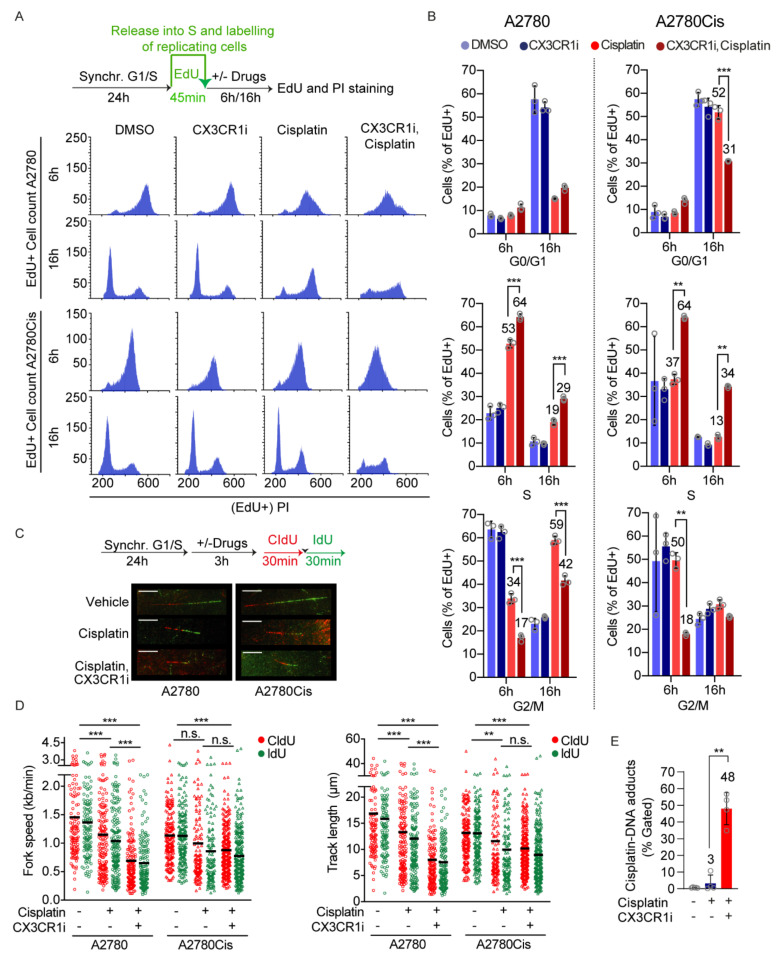
CX3CR1 promotes replicative S-phase repair of DNA crosslinks. (**A**,**B**) A2780 and A2780Cis cells were synchronized at the G1/S boundary for 24 h with 6 µM aphidicolin, released and pulsed with EdU for 45 min, and then treated with vehicle, 20 µM CDDP with or without 15 µM CX3CR1i for indicated timepoints. Cells were fixed with ethanol and stained for Edu and DNA by propidium iodide (PI) and subjected to flow cytometry. (**A**) Schematic of the experimental setup (top). Representative DNA histograms showing distribution of replicating cells in the G0/G1, S, and G2/M phases at indicated timepoints and treatments by first gating EdU positive cells followed by PI. (**B**) Data displayed as means ± SD, *n* = 3; ** *p*  <  0.01, *** *p*  <  0.001; two-way ANOVA, Tukey’s multiple comparisons test. (**C**,**D**) A2780 and A2780Cis cells were synchronized at the G1/S boundary as in (**A**), then released into vehicle, 2 µM (A2780) or 5 µM (A2780Cis) cisplatin with or without 10 µM CX3CR1i for 4 h with CldU (30 min) and IdU (30 min) added during the last hour. DNA fibers were prepared and visualized by immunofluorescence detection of CldU and IdU. (**C**) Representative images of DNA fibers and schematic of the experimental setup (top), red: CIdU, green: IdU, scale bar: 10 µm. (**D**) Data are displayed as scatter dot plots of average replication fork speed and average replication track lengths with means of *n* > 100 forks per condition. Shown is a representative experiment of *n* = 2, ** *p*  <  0.01, *** *p*  <  0.001, n.s. = nonsignificant; one-way ANOVA analysis, Tukey’s multiple comparisons test. (**E**) A2780 cells were synchronized at the G1/S border as in *(***A**) and released for 24 h into 10 µM cisplatin with or without 20 µM CX3CR1 prior fixation, cisplatin-DNA adducts were assessed by a cisplatin-DNA antibody in flow cytometry. Data displayed as means ± SD, *n* > 4; *** *p*  <  0.001; unpaired *t*-test, two-tailed.

**Figure 3 cancers-13-01442-f003:**
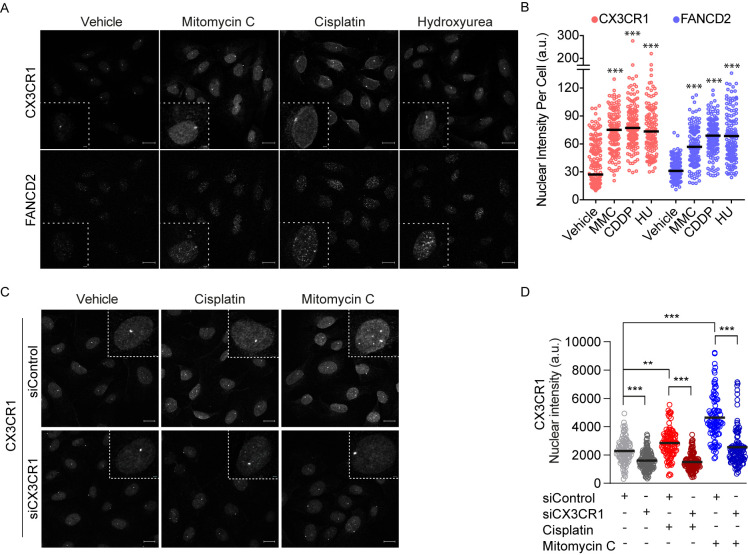
CX3CR1 accumulates in the nucleus upon FA pathway activation. (**A**,**B**) U2OS cells were treated for 24 h with 250 µM hydroxyurea (HU), 120 ng/mL mitomycin C (MMC), and 2.5 µM cisplatin (CDDP) prior to pre-extraction of soluble proteins and fixation. Cells were immunostained, nuclei stained with DAPI followed by confocal imaging. Images were analyzed using CellProfiler. (**A**) Representative confocal microscopy images for CX3CR1 and FANCD2 (left), dotted squares contain an enlargement of a selected representative cell per image, scale bar: 20 µm/2 µm. (**B**) Data displayed as scatter dot plot over indicated nuclear intensities with means of *n* > 100 cells per condition. Statistical significance is compared to vehicle, *** *p*  <  0.001; one-way ANOVA, Dunnett’s multiple comparisons test. (**C**,**D**) U2OS cells were transfected with indicated siRNAs for 48 h, then reseeded and synchronized for 24 h by aphidicolin treatments followed by release for 3 h into vehicle, 10 µM cisplatin, or 120 ng/mL mitomycin C prior to pre-extraction of soluble proteins and fixation. Cells were immunostained and nuclei stained with DAPI, subjected to confocal imaging and stainings quantified using CellProfiler. (**C**) Representative confocal images for CX3CR1, dotted lines indicate an enlargement of a selected cell per image, scale bar 20 µm/2 µm. (**D**) Scatter dot plots of CX3CR1 nuclear intensity of *n* > 100 cells per condition; *** *p*  <  0.001; one-way ANOVA, Tukey’s multiple comparisons test.

**Figure 4 cancers-13-01442-f004:**
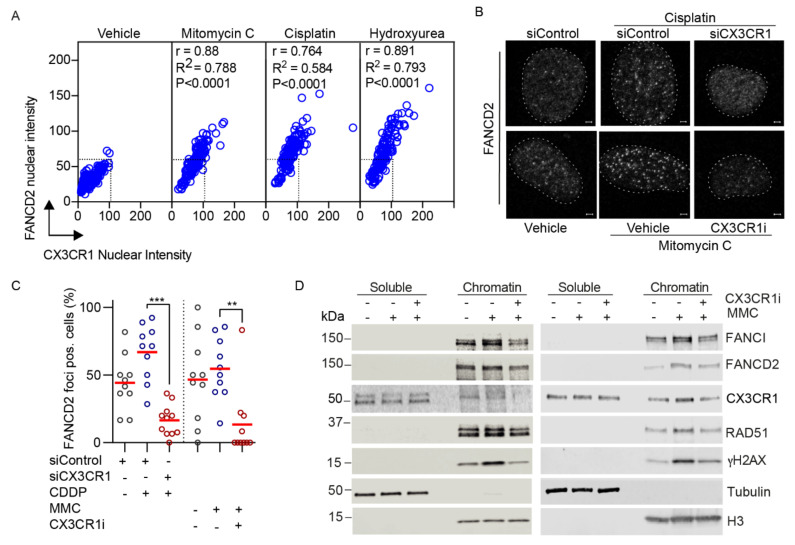
CX3CR1 modulates recruitment of key FA repair factors upon ICL induction. (**A**) Correlation plots of CX3CR1 and FANCD2 nuclear intensities from Figure 3A,B; Pearson correlation test. (**B**) Top panel, U2OS cells were transfected with siControl or siCX3CR1 siRNA pool for 48 h, then reseeded and synchronized for 24 h by aphidicolin treatments followed by release for 3 h into vehicle or 10 µM cisplatin prior to pre-extraction of soluble proteins, fixation, and immunostaining. Bottom panel, Aphidicolin synchronized U2OS cells were released in 120 ng/mL mitomycin C with or without 10 µM CX3CR1i for 3 h prior to pre-extraction of soluble proteins, fixation, and immunostaining. Nuclei were stained with DAPI. Shown are representative confocal images, dotted lines indicate nuclear border, scale bar 2 µm. (**C**) Scatter dot plot of FANCD2 foci positive cells, one data point indicates the percentage of foci positive cells per image, *n* > 100 cells per condition, red line indicates the mean; ** *p*  <  0.01, *** *p*  <  0.001; one-way ANOVA, Tukey’s multiple comparisons test. (**D**) A2780 (left panel) or U2OS (right panel) cells were synchronized as in (**B**), released for 3 h into vehicle, 120 ng/mL mitomycin C with or without 10 µM CX3CR1i followed by immunoblot of the soluble and nuclease-insoluble chromatin fractions, *n* > 3. Images of the uncropped Western blots can be found in Appendix A.

## Data Availability

Data is contained within the article or Appendix A or are available from the authors upon reasonable request.

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
