# Peer review of "Targeting CX3CR1 Suppresses the Fanconi Anemia DNA Repair Pathway and Synergizes with Platinum"

_cancers, 2021, doi:10.3390/cancers13061442_

Round 1

Reviewer 1 Report

In this manuscript, the authors reported a novel role for CX3CR1 in modulating the FA repair pathway by enabling FANCD2 recruitment at the stalled replication forks upon ICL induction. Interfering with CX3CR1 function upon ICL-induction led to inability of replicating cells to progress from S phase, replication fork stalling and impaired chromatin recruitment of key FA pathway factors. Further, CX3CR1i synergized with platinum agents in proliferation assays and reversed platinum resistance. Overall, the major conclusions can be supported by the data presented, several issues need to be clarified to improve the manuscript for publication.

Major:

  1. Fig. 1B-1D, to better support the conclusions that CX3CR1i can synergize with platinum agents in proliferation assays and reversed platinum resistance, all the proliferation assays should be performed using the A2780Cis cells as well. Moreover, according to Fig. S1, it is obvious that the knock-down efficiency of shCX3CR1#10 is much better than that of shCX3CR1#12 in the A2780 cells. Why the “colony formation assay” (Fig. 1C) was not done using cells transfected with shCX3CR1#10? All these must be addressed.
  2. Fig. 2E, the concentrations of cisplatin (10 µM) and CX3CR1i (20 µM) used in detecting the cisplatin-DNA adducts were at least 2-fold higher than those used in all other assays. This needs to be clarified. In addition, the measurement of cisplatin-DNA adducts in the A2780Cis cells should be also provided.
  3. It’s not clear why for all the mechanistic studies (Figs. 3 and 4), the authors switched the cell line to the U2OS cells. This must be clarified.
  4. Fig. S1, the protein expression level should be detected to validate the siCX3CR1 knockdown efficiency in the A2780 cells.

Minor:

  1. Fig.S3, line 27, “…CX3CR1-targeting siRNA oligo #4, # 5 and #6 in A2780 cells.” It should be U2OS cells instead.

Reviewer 2 Report

The manuscript titled “ Targeting CX3CR1 suppresses the Fanconi Anemia DNA repair pathway and reverses platinum resistance” describes the pharmacological study of a clinical phase small molecule inhibitor KAND567 of CX3CR1 on DNA crosslinker’s efficacy particularly in the platinum-resistant cells. The study demonstrates nicely CX3CR1 is involved in interstrand crosslink (ICL) repair using a genetic approach. It also shows that the CX3CR1 promotes ICL repair during S-phase. In general, the experiments are well designed with proper controls.

There a few concerns that need to be addressed before publishing

  • The protein level of CX3CR1 should be included to confirm the knockdown efficiency.

  • In Figure 3, it is difficult to see the CX3CR1 and FANCD2 immunostaining in the vehicle group in A. However, a much stronger immunostaining signal is shown in the panel C vehicle group even with siCX3CR1.

  • Where does the distinctive CX3CR1 nuclear body localize? it does not seem like they are damage-induced.
  • In Figure 4, the un-modified protein levels of FANCD2 and FANCI should be detectable in the soluble fraction. Also, the total lysate without fractionation should be presented to show the shift of ubiquitination upon treatment/

  • The CX3CR1 protein level should be included in figure 4D

  • Is the CS3CR1 localization also cell cycle-dependent? It would be nice to have a kinetic study of CX3CR1 translocation to the nucleus upon treatment of ICL agents and other DNA damaging agents.

Round 2

Reviewer 1 Report

All the concerns raised in the previous round review have been appropriately addressed by the authors. The manuscript is now suitable for publication.